# Natural diversity of the honey bee *(Apis mellifera)* gut bacteriome in various climatic and seasonal states

**Márton Papp**[1], **László Békési**[2], **Róbert Farkas**[2], **László Makrai**[3], **Maura Fiona Judge**[1], **Gergely Maróti**[4,5], **Dóra Tőzsér**[1], **Norbert Solymosi**[1] *

**1** Centre for Bioinformatics, University of Veterinary Medicine, Budapest, Hungary, **2** Department of Parasitology and Zoology, University of Veterinary Medicine, Budapest, Hungary, **3** Department of Microbiology and Infectious Diseases, University of Veterinary Medicine, Budapest, Hungary, **4** Plant Biology Institute of the Biological Research Center, Szeged, Hungary, **5** Faculty of Water Sciences, University of Public Service, Baja, Hungary

* solymosi.norbert@gmail.com

**Data Availability Statement:** The short read data of samples are publicly available and accessible through the PRJNA685398 from the NCBI Sequence Read Archive (SRA).

## Abstract

As pollinators and producers of numerous human-consumed products, honey bees have great ecological, economic and health importance. The composition of their bacteriota, for which the available knowledge is limited, is essential for their body's functioning. Based on our survey, we performed a metagenomic analysis of samples collected by repeated sampling. We used geolocations that represent the climatic types of the study area over two nutritionally extreme periods (March and May) of the collection season. Regarding bacteriome composition, a significant difference was found between the samples from March and May. The samples' bacteriome from March showed a significant composition difference between cooler and warmer regions. However, there were no significant bacteriome composition differences among the climatic classes of samples taken in May. Based on our results, one may conclude that the composition of healthy core bacteriomes in honey bees varies depending on the climatic and seasonal conditions. This is likely due to climatic factors and vegetation states determining the availability and nutrient content of flowering plants. The results of our study prove that in order to gain a thorough understanding of a microbiome's natural diversity, we need to obtain the necessary information from extreme ranges within the host's healthy state.

## Introduction

Honey bees are important pollinators with high economic value and ecosystem importance [1–3]. Their economic significance is based on their role in crop pollination and the different bee products they make [2, 3]. Honey, their most well-known product, is an important component of the human diet. Some evidence suggests that honey consumption can improve human health and might have a role in disease management [4]. However, honey bees are subjected to confined environments, and several factors threaten their health, including different

**Funding:** The project is supported (MP) by the European Union and co-financed by the European Social Fund (No. EFOP-3.6.3-VEKOP-16-2017-00005). It has also received funding (NS) from the European Union's Horizon 2020 research and innovation program under Grant Agreement No. 874735 (VEO). GM received support from the Hungarian Academy of Sciences through the Lendület-Programme (LP2020-5/2020). The funders had no role in study design, data collection and analysis, decision to publish, or preparation of the manuscript.

pathogens, parasites and chemicals used as pesticides in agriculture [5–7]. The global decline of this key pollinator poses a threat to food security and to the maintenance of biodiversity [8]. The composition of honey bee bacteriota, for which the available knowledge is limited, is essential for their body's functioning. While there is increasing attention on the effects of different herbicides and pathogens on the bee gut microbiota [9–11], only a few data are available on the natural variability of the microbiota. Nevertheless, this could form the basis of studies exploring the effects of different harmful agents on honey bees' gut bacteriota. Without this knowledge, one cannot decide if any suspected factor places the bacteriota composition into an adverse state. Although there are studies on honey bee gut microbiota and microbiomes [12–14], there is little evidence on the environmental factors affecting it. It is assumed that seasonal and environmental factors can have an influence on the gut bacteriome composition in honey bees, for example through feeding habits. During the collection season, various flowering plants provide diverse feed for the bees. The vegetation cycles, flowering and pollen quantity and quality of plants are mostly influenced by meteorological conditions, especially precipitation and temperature. Our study aimed to get more detailed knowledge about the natural variation of gut bacteriomes in healthy worker honey bees, based on seasonal and different environmental conditions, in a country-wide repeated measure survey. We have assumed that environmental factors will be associated with changes in the bacteriome as such changes were observed in other vertebrate [15, 16] and arthropod [17, 18] species. However, season showed contradictory results in honey bees, when observed in the honey producing season [19–21], although Kešnerová and colleagues [22] have found differences between the bacteriome of winter bees and foragers. To achieve our research goal, we were guided by the consideration that extreme states of the bee gut microbiome (environmentally and seasonally) should be sampled. Hence, samples were taken during the two most distinctive periods of the honey collection season and sampling sites were selected from markedly distinct areas based on their climatic characteristics.

## Materials and methods

### Sampling design and sample collection

The study's main goal was to understand the natural variability in the gut bacteriome of healthy honey bees (*Apis mellifera*). To measure seasonal variation, two sampling occasions were planned, one at the onset and one at the peak of the honey producing season. However, as season is not the only variable that could be considered when one is interested in the factors that could affect the bacteriome (climate could be an important environmental factor as well), we have determined our samples to be representative of Hungary on the climate level. To obtain such samples, we conducted a stratified spatial random sampling [23] as detailed below.

We gathered the 10 year average of the yearly growing degree days (GDD) with base 10 [24, 25] and the yearly total precipitation data for all the 175 local administrative units (level 1, hereafter refered to as LAU) in Hungary. Meteorological data for the period 2008–2017 was gathered from the ERA-Interim reanalysis data repository [26] by the spatial resolution of 0.125˚. We defined the two categories for our environmental variables as cooler-warmer and less-more for GDD and precipitation respectively.

Regarding GDD, the lower two quartiles were classified as cooler and the upper two quartiles as warmer. For precipitation, the yearly mean below the country-wide median was assumed as less and above the median as more. Each LAU was categorised by its own climatic variables (Fig 1). We created separate strata for each combinations of our two environmental variables.

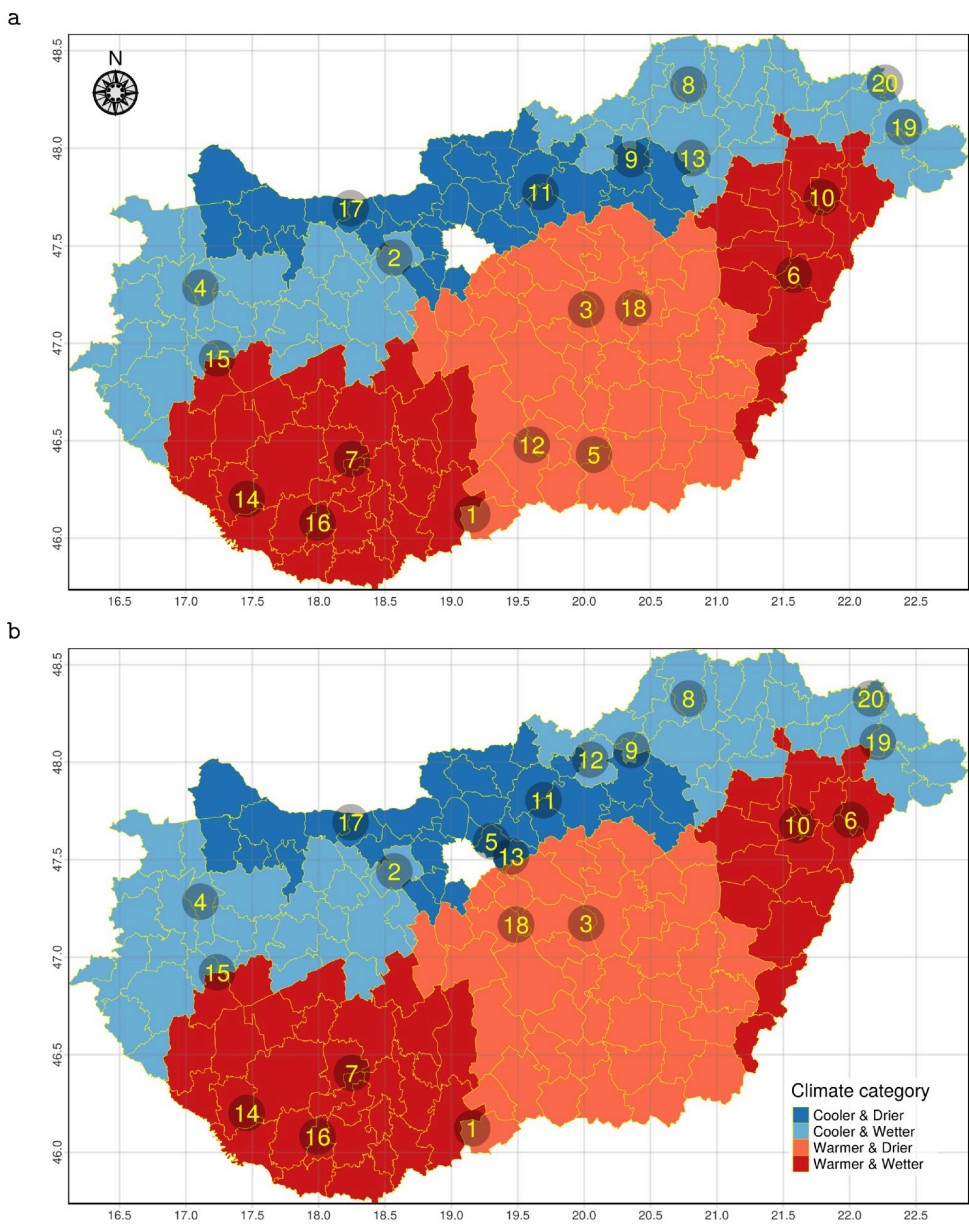

**Fig 1. Climate category spatial pattern and sampling points.** The Hungarian local administrative units (LAU) coloured by climatic categories based on growing degree days (GDD) and precipitation of the period 2008–2017. The numbers represent the identification numbers of the sampled apiaries in March (a) and May (b).

To ensure that our samples are representative of Hungary at the climatic condition level, we have selected 20 LAUs so that the sample size of each previously defined strata was proportional to the stratifying GDD and precipitation categories' country-wide frequency. The R [27] package spsurvey [23, 28] was used for the stratified spatial random sampling of the LAUs as described above. One apiary was selected from each appointed LAU (making the total number of selected apiaries 20). To minimise the effect of the keeping conditions on our results, each apiary was selected based on personal conversations. Since in Hungary mainly Carniolan honey bees (*Apis mellifera carnica*) are in operation, the samples were drawn from colonies of that subspecies.

Sample collection was performed twice during the honey producing season (at the onset and at the peak). The first was done between 20/03/2019 and 25/03/2019 (Fig 1a) and the second during the period of 23/05/2019 to 01/06/2019 (Fig 1b), hereafter referred to as the sampling periods of March and May respectively. To obtain a representative sample of workers, two-level pooling was performed by apiaries to reduce the effects of possible biasing factors (e.g. age heterogeneity).

Three colonies were selected for sampling (in each apiary) at the first sample collection (sampling period March) and these same colonies were sampled at the second sample collection (sampling period May). Each time 20 workers were collected and frozen immediately by dry ice from each of the three selected colonies from every apiary.

During the sampling in May, in two apiaries (ID: 6, 13) only two of the three colonies from the March sampling period were accessible so only these two were sampled. Migration of the colonies occurred in eight of the apiaries (ID: 5, 6, 9, 10, 12, 13, 18, 20) between the two sampling periods. In the case of four apiaries of the above eight (ID: 5, 9, 12, 13), the environmental classification has changed between the two samplings. Two apiaries moved from warmer to cooler regions (ID 5, 12) and ID 12 apiary along with ID 9 moved from a LAU with less precipitation to one with more. One apiary (ID 13) has migrated from a region with more precipitation to a region with less. The sample sizes used by apiaries and sampling period are summarized in S1 Table.

Data on animal health history was collected in each of the sampled herds at both sampling times by questionnaire. We asked if the beekeeper had experienced significant mortality in the previous season or overwintering by the March sampling. By the May sampling, we asked if there was significant mortality between the two sampling times. In each of the apiaries at both sampling times, we received the answer that no such events were experienced.

## Sample preparation

The collected samples were prepared for next-generation sequencing (NGS) in the Department of Parasitology and Zoology, University of Veterinary Medicine Budapest. From the deep-frozen workers, 10 by colonies were chosen. The bees' entire gastrointestinal tracts were removed and pooled on the apiary-sampling level. The gut preparation forceps was never used before and one forceps was applied only for one pool (3x10 guts) processing.

## DNA extraction and metagenomics library preparation

The Quick-DNA Fecal/Soil Microbe Kit from Zymo Research was used for the simple and rapid isolation of inhibitor-free, high-quality host cell and microbial DNA from the bee gut samples. Isolated total metagenome DNA was used for library preparation. In vitro fragment libraries were prepared using the NEBNext Ultra II DNA Library Prep Kit for Illumina. Paired-end fragment reads were generated on an Illumina NextSeq sequencer using TG Next-Seq 500/550 High Output Kit v2 (300 cycles). Primary data analysis (base-calling) was carried out with Bbcl2fastq software (v2.17.1.14, Illumina).

## Bioinformatic analysis

After merging the paired-end reads by PEAR [29], quality-based filtering and trimming was performed by Adapterremoval [30] using 15 as the quality threshold and only retaining reads longer than 50 bp. The *Apis mellifera* genome (Amel_HAv3.1) sequences host contaminants were filtered out by Bowtie2 [31] with the very-sensitive-local setting minimizing the false positive match level [32] in further metagenome classification. The remaining reads, after deduplication by VSEARCH [33], were taxonomically classified using Kraken2 (k = 35) [34] with the

NCBI non-redundant nucleotide database [35]. The core bacteria was defined as the relative abundance of agglomerated counts on species-level above 0.1% in at least half of the samples. The taxon classification data was managed in R [27] using functions of package phyloseq [36] and microbiome [37].

## Statistical analysis

The within-subject diversity ($\alpha$-diversity) was assessed using the numbers of observed species (richness) and the Inverse Simpson's Index (evenness). These indices were calculated in 1,000 iterations of rarefied OTU tables with a sequencing depth of 6,129. The average over the iterations was taken for each apiary. The $\alpha$-diversity expressed by Inverse Simpson's Index was compared between the conditions using linear models. Comparing the samples collected in March and May, a mixed-effect model was applied to handle the repeated measure by apiary as a random factor. The between-subject diversity ($\beta$-diversity) was assessed by Bray-Curtis distance [38] based on the relative abundances of bacterial species. Using this measure, non-metric multidimensional scaling (NMDS) ordination was applied to visualise the samples' dissimilarity. To examine statistically whether the bacterial species composition differed by climatic or seasonal conditions, PERMANOVA (Permutational Multivariate Analysis of Variance [39]) and PERMDISP2 [40] procedures were performed using vegan package [41] in R [27]. The abundance differences in the core bacteriome according to the seasonal and climatic conditions were analysed by a negative binomial generalised model of DESeq2 package [42] in R [27]. This approach was applied following the recommendation of Weiss et al. [43]. None of the compared groups had more than 20 samples and their average library size ratio was less than 10. Since the apiaries were sampled repeatedly for capturing the seasonal effect, the samples were paired in the model. Regarding the multiple comparisons, an FDR-adjusted p-value (q-value) less than 0.10 was considered significant. The statistical tests were two-sided.

## Results

In the results of our study, we first summarize the most relevant indicators of sequencing and taxon classification. After the within- and between-subject diversity of the whole bacteriome, we present the differentiating species of the core bacteriome.

## Sequencing and taxon classification

The shotgun sequencing generated paired-end read counts of samples ranged between 311,931 and 546,924 with a mean of 413,629. The OTU table, created by Kraken2 taxonomic classification, contained counts of samples ranging between 11,646 and 114,573 with a mean of 44,280. The minimum, maximum and median read counts of the samples assigned as bacterial species were 6,129, 62,836 and 270,774 respectively.

## Within-subject diversity

The numbers of observed species and the Inverse Simpson's Index $\alpha$-diversity metrics by environmental and seasonal strata are shown in Fig 2. The Inverse Simpson's Index outliers in the samples collected in March from districts with less and more precipitation are the apiary ID 9 and ID 13 respectively. The apiary ID 12 sampled in March had an outlying high number of observed species too. From the same sampling period among the samples gathered from districts with more precipitation, apiary ID 8 appears to be an outlier.

In samples from the cooler environment collected in March, the $\alpha$-diversity was significantly (p = 0.0215) higher than in samples from warmer districts. There was no significant

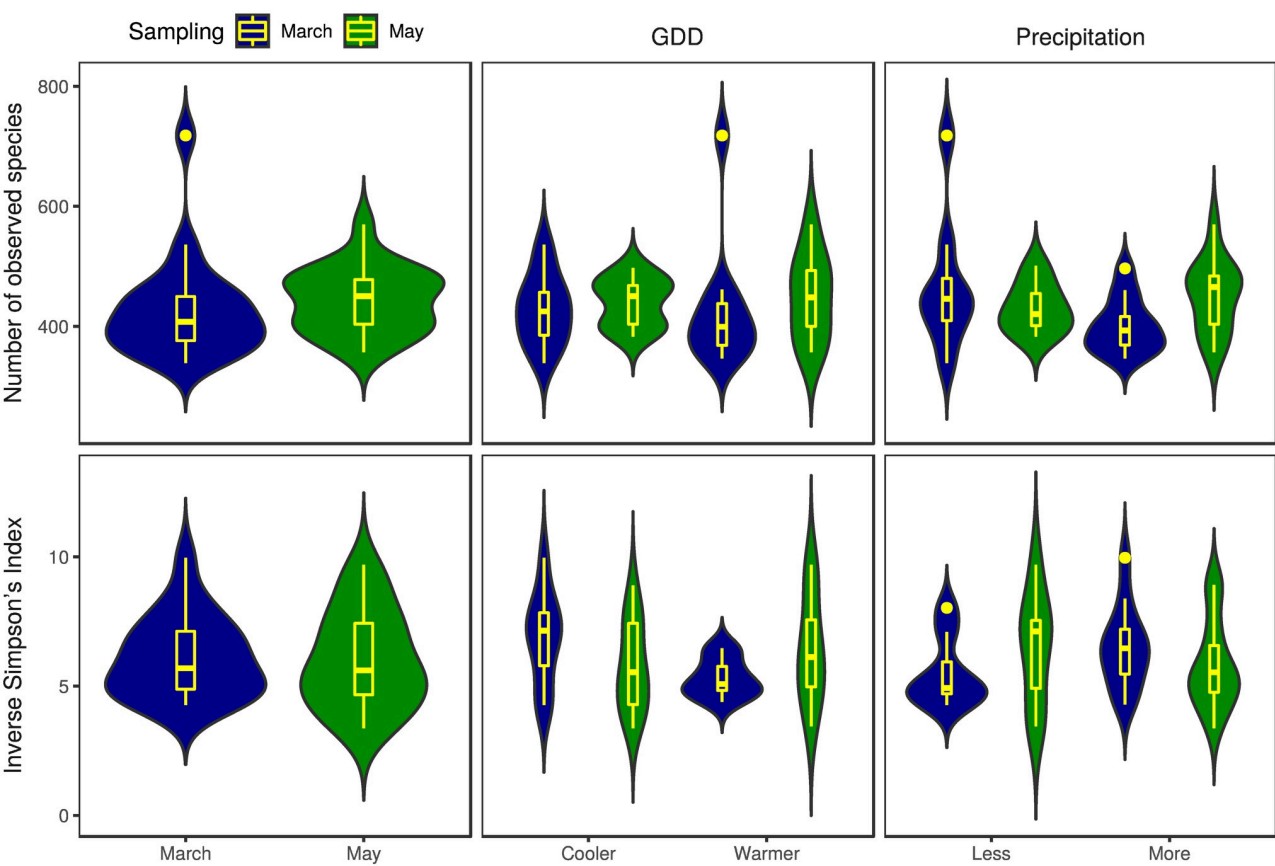

**Fig 2. Richness and evenness of honey bee gut bacteriome by sample groups.** The numbers of observed species (richness) and the Inverse Simpson's Index (evenness) as $\alpha$-diversity metrics are presented as a violin and box plot combination. These indices were calculated in 1,000 iterations of rarefied OTU tables with a sequencing depth of 6,129. The average over the iterations was taken for each apiary. The violin plot shows the probability density, while the box plot marks the outliers, median and the IQR. For Inverse Simpson's Index, the comparison of samples from cooler and warmer districts collected in March showed significant (p = 0.0215) differences.

difference in $\alpha$-diversity between the precipitation categories of samples collected in March (p = 0.178). In samples collected in May, there was no significant difference between GDD or precipitation categories (p = 0.463 and p = 0.456 respectively).

## Between-subject diversity

The dissimilarity of the samples' bacterial species profiles ($\beta$-diversity) is visualised by NMDS ordination (Figs 3 and 4) based on Bray-Curtis distance. The ordination stress was 0.144, 0.062 and 0.116 for all samples, samples of March and samples of May respectively. By PERMA-NOVA analysis of bacterial species composition, a significant (p = 0.002) difference was found between the samples from March and May. The samples' bacteriome from March showed a similar significant (p = 0.02) distance between the cooler and the warmer districts. From the same period, the precipitation levels did not differ significantly (p = 0.155). In the samples gathered in May, there was no significant distance between GDD and precipitation categories (p = 0.277 and p = 0.849 respectively). Both significant PERMANOVA results were confirmed by the PERMDISP2 distance-based tests for homogeneity of multivariate dispersions (p = 0.033 and p = 0.003 respectively).

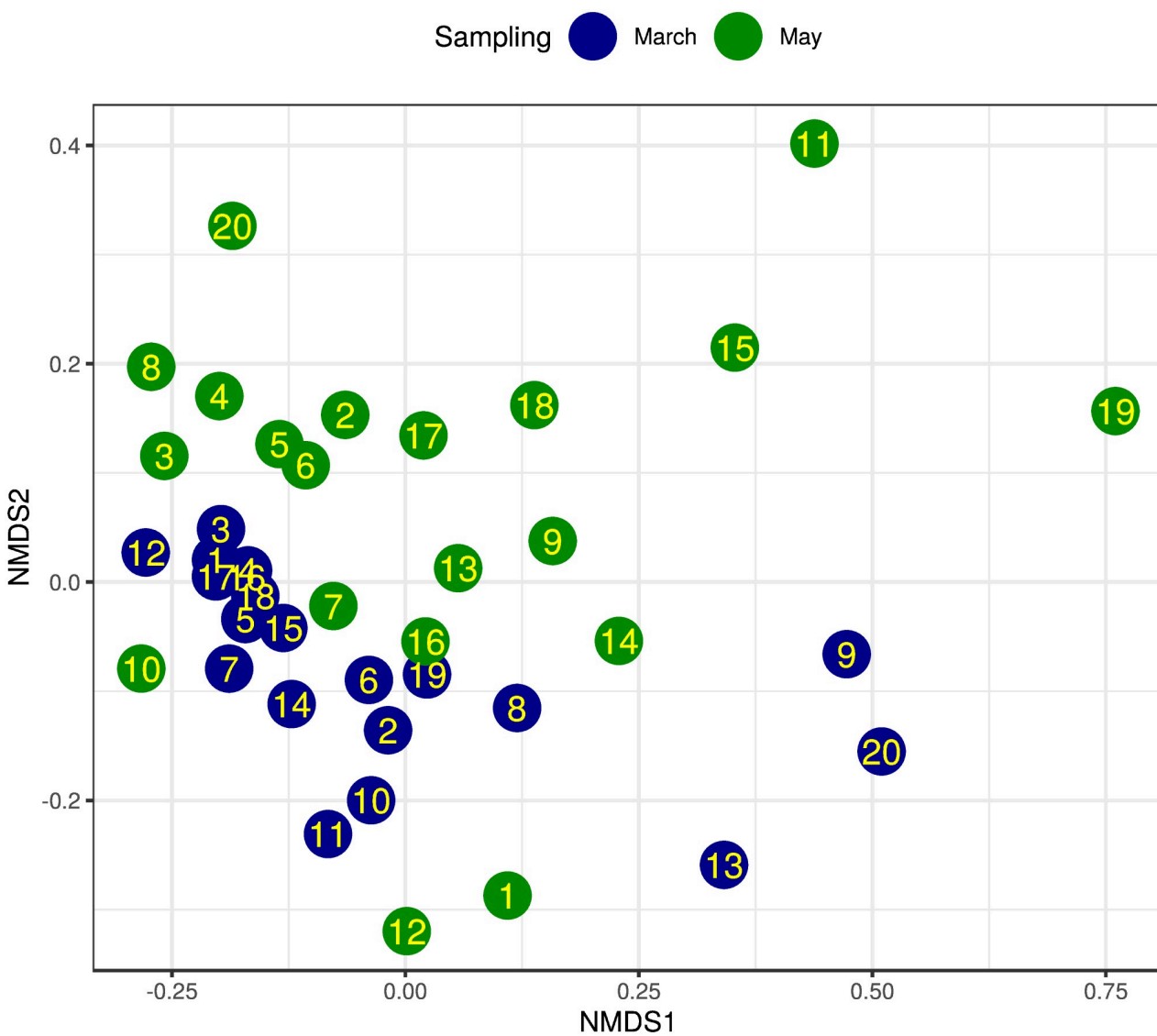

**Fig 3. NMDS ordination of bacteriome for sampling March and May.** Bray-Curtis dissimilarity was calculated using the species-level abundance of core bacteria. The samples from apiaries (IDs in dots) collected in March (blue) and May (green) are plotted using these dissimilarities. Based on the same measures, PERMANOVA analysis showed significant differences between the sampling time periods (p = 0.002, stress = 0.144).

## Core bacteriome and differentiating species

The core bacteriome members having relative abundance above 0.1% in at least half of the samples are *Bartonella apis*, *Bifidobacterium asteroides*, *Bifidobacterium coryneforme*, *Bifidobacterium indicum*, *Commensalibacter* sp. AMU001, *Frischella perrara*, *Gilliamella apicola*, *Lactobacillus apis*, *Lactobacillus bombi*, *Lactobacillus helsingborgensis*, *Lactobacillus kullabergensis*, *Lactobacillus kunkeei*, *Lactobacillus mellis*, *Lactobacillus* sp. wkB8 and *Snodgrassella alvi*. The relative abundances of each apiary's core bacteriome species are plotted by sampling periods and environmental strata in Fig 5. Table 1 shows the overall and grouped mean and standard deviation of core bacteriome species' relative abundances.

Associations between seasonal conditions, climatic condition levels and the abundance of core bacteriome species were examined using negative binomial generalized linear models

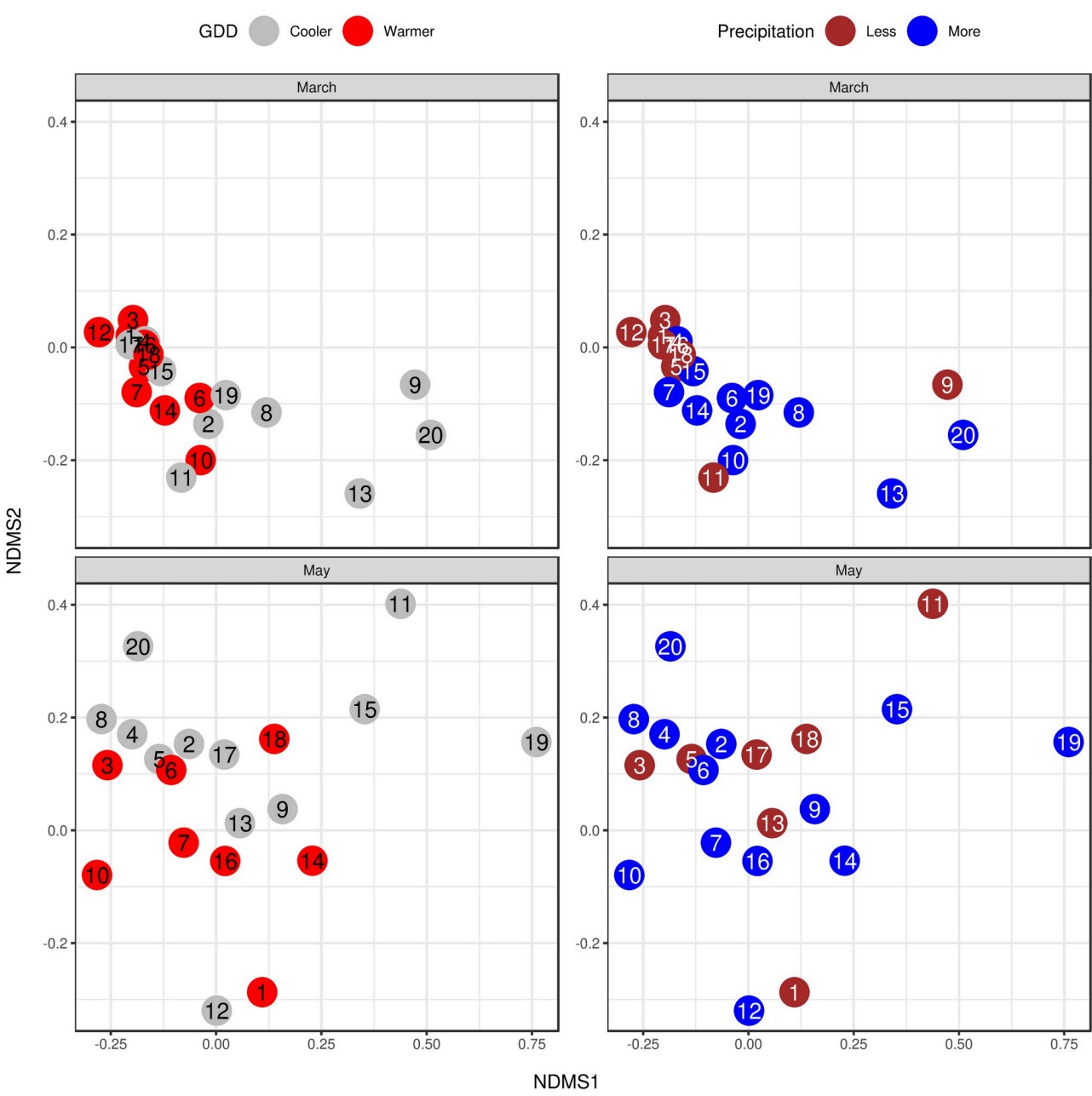

**Fig 4. NMDS ordination of bacteriome for environmental condition categories by sampling period.** The colours represent the environmental condition categories and the numbers correspond to the apiary IDs. The stress was 0.062 and 0.116 for March and May respectively. The samples' bacteriome from March showed significant (p = 0.02) distance between the cooler and warmer districts. From the same period, the precipitation levels did not differ significantly (p = 0.155). In the samples gathered in May, there was no significant distance neither between GDD nor precipitation categories (p = 0.277 and p = 0.849, respectively).

[42] (Table 2). The abundance of *B. apis* (FC: 15.41, q<0.00001), *B. asteroides* (FC: 1.61, q = 0.0084), *C.* sp. AMU001 (FC: 2.46, q = 0.00001), *L. helsingborgensis* (FC: 1.7, q = 0.008) and *S. alvi* (FC: 1.49, q = 0.011) significantly increased from March to May. In the same comparison, the abundance of *L. apis* (FC: 0.64, q = 0.0066), *L. bombi* (FC: 0.64, q = 0.0052), *L. kullabergensis* (FC: 0.57, q = 0.00056) and *L. mellis* (FC: 0.64, q = 0.0052) was significantly decreased. In the samples collected in March, the abundance of *L. kunkeei* (FC: 3.86, q = 0.094)

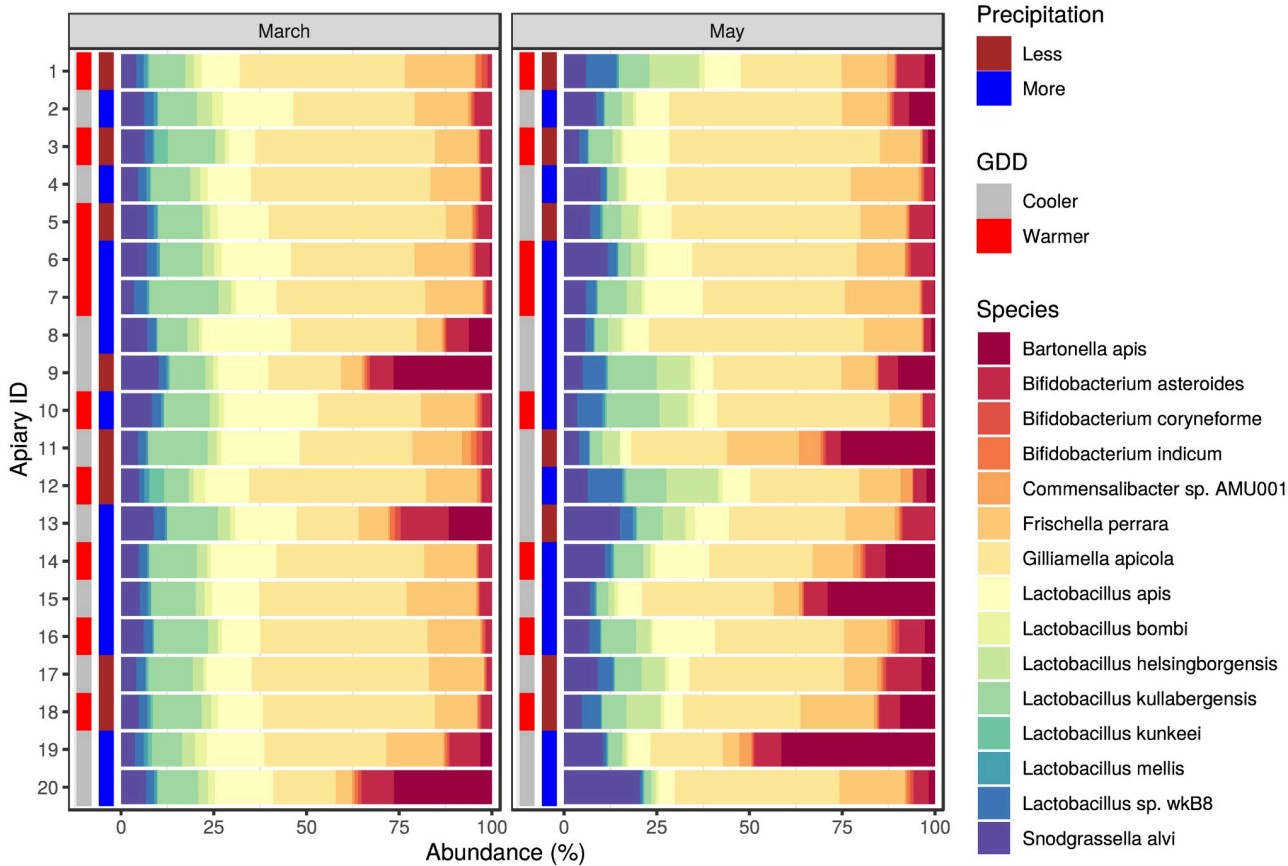

**Fig 5. Core bacteriome composition of honey bee gut samples.** The relative abundance is plotted for the first (March) and second (May) sampling. Besides the bacterial species of the core bacteriome, the environmental condition (growing degree-day (GDD) and precipitation) categories of sampling places are also marked.

was significantly higher in warmer regions than in cooler ones. In the same period, the abundance of *B. apis* (FC: 0.02, q<0.00001) and *B. asteroides* (FC: 0.47, q = 0.0027) was significantly lower in warmer LMUs than in cooler ones. In March samples, the abundance of *L. kunkeei* (FC: 0.13, q = 0.011) was significantly lower in districts with more precipitation than in LMUs with less precipitation. In samples collected in May, none of the core bacteriome species showed significant alterations in abundance neither by GDD categories nor by the precipitation levels. The relative abundance distribution of significantly different species per group is summarized in S1 Fig.

## Discussion

Bees as pollinators are essential for both ecology and agriculture. Several results have been reported on the composition of their gut bacteria. However, repeated measurement results from climatic stratified spatial sampling on a country-wide basis are not known in the literature. Our work aimed to gain data regarding the natural diversity of gut bacteriota in healthy worker bees. For this objective, we classified the local administrative units of Hungary into climatic strata based on the longer time series commonly used in climatology. In climate science, this is used to filter out weather fluctuations, thus determining the climate of an area for a given period. We compared warmer to colder and drier to wetter areas within the March

**Table 1. Relative abundances by environmental and seasonal categories.**

| | Species | All samples Mean (SD) | GDD Cooler | Warmer | Precipitation Less | More |
|---|---|---|---|---|---|---|
| March | | | | | | |
| | *Bartonella apis* | 3.78 (8.25) | 7.41 (10.7) | 0.14 (0.14) | 3.41 (9.32) | 4.02 (7.89) |
| | *Bifidobacterium asteroides* | 4.08 (3.02) | 5.64 (3.62) | 2.52 (0.92) | 2.83 (1.64) | 4.92 (3.49) |
| | *Bifidobacterium coryneforme* | 0.64 (0.43) | 0.72 (0.48) | 0.56 (0.38) | 0.73 (0.54) | 0.58 (0.35) |
| | *Bifidobacterium indicum* | 0.62 (0.41) | 0.69 (0.44) | 0.55 (0.39) | 0.71 (0.52) | 0.56 (0.33) |
| | *Commensalibacter* sp. AMU001 | 0.56 (0.49) | 0.66 (0.65) | 0.45 (0.24) | 0.7 (0.73) | 0.46 (0.22) |
| | *Frischella perrara* | 12.49 (4.11) | 11.46 (4.85) | 13.52 (3.13) | 12 (4.25) | 12.82 (4.17) |
| | *Gilliamella apicola* | 37 (10.63) | 31.92 (11.56) | 42.08 (6.9) | 41.6 (10.67) | 33.93 (9.85) |
| | *Lactobacillus apis* | 15.05 (4.73) | 16.23 (4.03) | 13.87 (5.29) | 12.76 (3.93) | 16.57 (4.75) |
| | *Lactobacillus bombi* | 1.73 (0.71) | 1.75 (0.78) | 1.71 (0.68) | 1.72 (0.75) | 1.74 (0.72) |
| | *Lactobacillus helsingborgensis* | 2.71 (0.61) | 2.91 (0.57) | 2.52 (0.61) | 2.26 (0.49) | 3.01 (0.49) |
| | *Lactobacillus kullabergensis* | 11.78 (2.77) | 11.17 (2.42) | 12.4 (3.08) | 11.49 (2.82) | 11.98 (2.84) |
| | *Lactobacillus kunkeei* | 0.6 (1.13) | 0.27 (0.35) | 0.94 (1.52) | 1.22 (1.61) | 0.2 (0.3) |
| | *Lactobacillus mellis* | 0.75 (0.29) | 0.72 (0.29) | 0.77 (0.31) | 0.77 (0.36) | 0.73 (0.26) |
| | *Lactobacillus* sp. wkB8 | 2.34 (0.41) | 2.37 (0.28) | 2.3 (0.53) | 2.07 (0.34) | 2.52 (0.37) |
| | *Snodgrassella alvi* | 5.88 (1.8) | 6.09 (2.11) | 5.66 (1.5) | 5.74 (2.06) | 5.97 (1.69) |
| May | | | | | | |
| | *Bartonella apis* | 7.6 (11.43) | 10.13 (13.9) | 3.81 (4.87) | 6.19 (8.99) | 8.36 (12.84) |
| | *Bifidobacterium asteroides* | 5.12 (2.24) | 5.29 (2.37) | 4.86 (2.16) | 6.11 (2.83) | 4.59 (1.75) |
| | *Bifidobacterium coryneforme* | 0.43 (0.29) | 0.42 (0.28) | 0.44 (0.33) | 0.42 (0.2) | 0.43 (0.33) |
| | *Bifidobacterium indicum* | 0.44 (0.28) | 0.43 (0.25) | 0.45 (0.34) | 0.41 (0.19) | 0.45 (0.33) |
| | *Commensalibacter* sp. AMU001 | 1.33 (1.37) | 1.58 (1.66) | 0.96 (0.69) | 1.72 (1.89) | 1.13 (1.02) |
| | *Frischella perrara* | 12.79 (4.43) | 12.42 (4.67) | 13.35 (4.28) | 13.79 (4.18) | 12.25 (4.62) |
| | *Gilliamella apicola* | 38.71 (10.71) | 38.92 (11.45) | 38.4 (10.25) | 37.91 (12.12) | 39.15 (10.38) |
| | *Lactobacillus apis* | 8.74 (3.92) | 6.82 (2.18) | 11.62 (4.29) | 7.67 (3.26) | 9.32 (4.24) |
| | *Lactobacillus bombi* | 0.97 (0.5) | 0.95 (0.56) | 1 (0.43) | 1.07 (0.73) | 0.92 (0.34) |
| | *Lactobacillus helsingborgensis* | 5.15 (3.81) | 4.74 (3.79) | 5.77 (4) | 6.64 (3.69) | 4.35 (3.76) |
| | *Lactobacillus kullabergensis* | 6.52 (3.43) | 5.56 (3.42) | 7.96 (3.08) | 6.26 (1.61) | 6.66 (4.15) |
| | *Lactobacillus kunkeei* | 0.18 (0.17) | 0.18 (0.2) | 0.18 (0.14) | 0.12 (0.15) | 0.21 (0.18) |
| | *Lactobacillus mellis* | 0.42 (0.19) | 0.41 (0.2) | 0.42 (0.19) | 0.43 (0.27) | 0.4 (0.15) |
| | *Lactobacillus* sp. wkB8 | 3.44 (2.44) | 3.07 (2.5) | 4.01 (2.4) | 4.06 (2.07) | 3.11 (2.63) |
| | *Snodgrassella alvi* | 8.15 (4.16) | 9.09 (4.64) | 6.74 (3.07) | 7.18 (3.91) | 8.67 (4.35) |

samples. Samples taken in May were analyzed in the same way. March and May samples from the same apiaries were also compared. Thus, our results provide new data on the effects of the climatic environment and the seasons. As some stocks changed their location during the two samplings, the question may arise as to how this may bias the results. In our opinion, this effect is negligible, as the climatic strata were compared within the March samples and within the May samples. Most can be suggested that the May samples are not climatically representative since, in the case of four apiaries, the environment in May was different from that in March.

We have evaluated differences between environmental conditions, namely temperature and precipitation. It is well known that honey bees possess a relatively simple bacteriome in their gut, constituted of only a small number of species. There are members of the bacteriome that are always present, these are often referred to as core members. Core bacteria of the honey bee gut bacteriome are *S. alvi*, *G. apicola*, and a few *Lactobacillus* and *Bifidobacterium* species. Besides the core members, there are frequent but not essential species as well, e.g. *B. apis* and *F. perrara* [17, 19, 44–46]. Our results are in agreement with these previous findings, as the

**Table 2. Abundance alterations of core bacteriome by seasonal and climatic conditions.** A negative binomial model estimated the association between species abundance of core bacteriome and sampling seasons, GDD- and precipitation level.

| Samples | Species | Mean counts[†] | Fold change (95% CI) | q [§] | Mean counts | Fold change (95% CI) | q |
|---|---|---|---|---|---|---|---|
| | | GDD Warmer vs. cooler | | | Precipitation More vs. less | | |
| May | *Bartonella apis* | 1068.87 | 0.02 (0, 0.08) | <0.00001 | 1068.87 | 0.99 (0.15, 6.53) | * |
| | *Bifidobacterium asteroides* | 1144.10 | 0.47 (0.31, 0.71) | 0.00274 | 1144.10 | 1.52 (0.94, 2.47) | 0.34335 |
| | *Bifidobacterium coryneforme* | 180.26 | 0.78 (0.47, 1.3) | 0.55291 | 180.26 | 0.72 (0.43, 1.2) | 0.44555 |
| | *Bifidobacterium indicum* | 176.76 | 0.81 (0.49, 1.33) | 0.55291 | 176.76 | 0.72 (0.43, 1.19) | 0.44555 |
| | *Commensalibacter* sp. AMU001 | 162.07 | 0.68 (0.37, 1.25) | 0.46158 | 162.07 | 0.58 (0.32, 1.05) | 0.34335 |
| | *Frischella perrara* | 3662.30 | 1.18 (0.84, 1.65) | 0.55291 | 3662.30 | 0.97 (0.68, 1.37) | 0.85067 |
| | *Gilliamella apicola* | 11004.31 | 1.29 (0.94, 1.78) | 0.42005 | 11004.31 | 0.75 (0.55, 1.04) | 0.34335 |
| | *Lactobacillus apis* | 4366.20 | 0.84 (0.65, 1.09) | 0.46158 | 4366.20 | 1.17 (0.89, 1.54) | 0.47268 |
| | *Lactobacillus bombi* | 495.19 | 1.03 (0.73, 1.44) | 0.86802 | 495.19 | 0.89 (0.63, 1.26) | 0.67884 |
| | *Lactobacillus helsingborgensis* | 786.53 | 0.87 (0.71, 1.07) | 0.46158 | 786.53 | 1.2 (0.96, 1.5) | 0.34335 |
| | *Lactobacillus kullabergensis* | 3454.42 | 1.11 (0.87, 1.41) | 0.55291 | 3454.42 | 0.96 (0.74, 1.24) | 0.80541 |
| | *Lactobacillus kunkeei* | 187.49 | 3.86 (1.25, 11.89) | 0.09368 | 187.49 | 0.13 (0.05, 0.36) | 0.01120 |
| | *Lactobacillus mellis* | 213.99 | 1.13 (0.79, 1.61) | 0.62292 | 213.99 | 0.86 (0.59, 1.25) | 0.59928 |
| | *Lactobacillus* sp. wkB8 | 681.21 | 0.97 (0.8, 1.18) | 0.84023 | 681.21 | 1.1 (0.89, 1.36) | 0.58331 |
| | *Snodgrassella alvi* | 1706.23 | 0.92 (0.71, 1.2) | 0.62292 | 1706.23 | 0.94 (0.71, 1.24) | 0.78937 |
| May | *Bartonella apis* | 1553.31 | 0.28 (0.08, 1.01) | 0.22572 | 1553.31 | 1.46 (0.36, 5.85) | 0.86672 |
| | *Bifidobacterium asteroides* | 942.33 | 0.83 (0.56, 1.22) | 0.74756 | 942.33 | 0.9 (0.6, 1.37) | 0.86672 |
| | *Bifidobacterium coryneforme* | 79.75 | 0.92 (0.49, 1.73) | 0.98269 | 79.75 | 1.13 (0.59, 2.18) | 0.86672 |
| | *Bifidobacterium indicum* | 81.39 | 0.9 (0.49, 1.63) | 0.98269 | 81.39 | 1.23 (0.66, 2.28) | 0.86672 |
| | *Commensalibacter* sp. AMU001 | 255.76 | 0.51 (0.26, 1) | 0.22572 | 255.76 | 0.72 (0.35, 1.49) | 0.86672 |
| | *Frischella perrara* | 2522.61 | 1.02 (0.66, 1.56) | 0.98269 | 2522.61 | 0.98 (0.62, 1.54) | 0.92494 |
| | *Gilliamella apicola* | 7869.43 | 1.01 (0.65, 1.57) | 0.98269 | 7869.43 | 1.09 (0.69, 1.73) | 0.86672 |
| | *Lactobacillus apis* | 1698.58 | 1.62 (1.07, 2.45) | 0.22572 | 1698.58 | 1.23 (0.77, 1.97) | 0.86672 |
| | *Lactobacillus bombi* | 183.30 | 1.06 (0.68, 1.66) | 0.98269 | 183.30 | 1.03 (0.65, 1.65) | 0.92494 |
| | *Lactobacillus helsingborgensis* | 952.28 | 1.2 (0.66, 2.17) | 0.98269 | 952.28 | 0.79 (0.43, 1.45) | 0.86672 |
| | *Lactobacillus kullabergensis* | 1258.90 | 1.5 (0.93, 2.41) | 0.29310 | 1258.90 | 1.17 (0.7, 1.97) | 0.86672 |
| | *Lactobacillus kunkeei* | 36.06 | 1.08 (0.39, 2.97) | 0.98269 | 36.06 | 1.65 (0.59, 4.57) | 0.86672 |
| | *Lactobacillus mellis* | 78.82 | 1 (0.65, 1.56) | 0.98269 | 78.82 | 1.12 (0.7, 1.77) | 0.86672 |
| | *Lactobacillus* sp. wkB8 | 648.44 | 1.33 (0.75, 2.36) | 0.74756 | 648.44 | 0.91 (0.49, 1.66) | 0.86672 |
| | *Snodgrassella alvi* | 1567.32 | 0.66 (0.43, 1.02) | 0.22572 | 1567.32 | 1.43 (0.9, 2.25) | 0.86672 |
| | | May vs. March | | | | | |
| All | *Bartonella apis* | 1538.45 | 15.41 (6.07, 39.17) | <0.00001 | | | |
| | *Bifidobacterium asteroides* | 1074.14 | 1.61 (1.16, 2.24) | 0.00837 | | | |
| | *Bifidobacterium coryneforme* | 118.29 | 0.74 (0.52, 1.05) | 0.12117 | | | |
| | *Bifidobacterium indicum* | 118.69 | 0.78 (0.55, 1.11) | 0.18588 | | | |
| | *Commensalibacter* sp. AMU001 | 232.29 | 2.46 (1.7, 3.57) | 0.00001 | | | |
| | *Frischella perrara* | 3046.15 | 1.17 (0.85, 1.61) | 0.34279 | | | |
| | *Gilliamella apicola* | 9304.64 | 1.22 (0.92, 1.62) | 0.18588 | | | |
| | *Lactobacillus apis* | 2740.93 | 0.64 (0.48, 0.86) | 0.00656 | | | |
| | *Lactobacillus bombi* | 309.86 | 0.64 (0.48, 0.84) | 0.00519 | | | |
| | *Lactobacillus helsingborgensis* | 928.22 | 1.7 (1.19, 2.43) | 0.00800 | | | |
| | *Lactobacillus kullabergensis* | 2115.13 | 0.57 (0.43, 0.76) | 0.00056 | | | |
| | *Lactobacillus kunkeei* | 99.88 | 0.64 (0.36, 1.14) | 0.16276 | | | |
| | *Lactobacillus mellis* | 133.69 | 0.64 (0.49, 0.85) | 0.00519 | | | |

(*Continued*)

**Table 2.** (Continued)

| Samples | Species | Mean counts[†] | Fold change (95% CI) | $q$ [§] | Mean counts | Fold change (95% CI) | $q$ |
|---------|---------|------------|----------------------|------|-------------|----------------------|-----|
| | | **GDD** **Warmer vs. cooler** | | | **Precipitation** **More vs. less** | | |
| | *Lactobacillus* sp. wkB8 | 680.91 | 1.35 (0.98, 1.87) | 0.10342 | | | |
| | *Snodgrassella alvi* | 1700.90 | 1.49 (1.12, 1.98) | 0.01099 | | | |

[†]Sequence read counts were normalized by dividing raw counts by DESeq size factors

[§]FDR-adjusted p-value. FDR adjustment was conducted in each pairwise comparison separately

[*]DESeq method can't estimate p-values without outlier replacement

core bacteriome found in this study contained *Snodgrasella*, *Gilliamella* and different *Lactobacillus* and *Bifidobacterium* species (*B. asteroides*, *B. coryneforme*, *B. indicum*, *L. apis*, *L. helsinborgensis*, *L. kullabergensis*, *L. mellis*) which were all previously described as core members. Also, the frequently observed but non-core members *B. apicola* and *F. perrara* were also present. Surprisingly, we found *L. kunkeei* to be present in every sample in our experiment, which, although being a regular member in the honey crop, rarely presents in the gut [21]. The presence of *L. kunkeei* in the core bacteriome of our study could be the result of our gut preparation protocol, namely that the whole gastrointestinal tract was extracted during sample processing.

We assumed that a seasonal shift in gut bacteriome would be identified as such differences have been observed in many other invertebrate and vertebrate species. For instance, seasonal variation of the gut bacteriome was found in humans [47, 48], non-human primates [49, 50], other mammalian species [51, 52], fishes [15], birds [53] and arthropods as well [54, 55]. However, seasonal changes are most likely linked to other factors, such as changes in the feeding habit of the animal or its lifestyle. Thus other potential factors, such as the effect of environmental conditions, should also be accounted for when the natural variation of the microbiome is considered. Although there is a lot of information on the honey bee gut bacteriome composition, little is known about its seasonal and environmental variation. Kešnerová and colleagues [22] examined the variation of the honey bee gut bacteriome throughout a year and found marked differences between winter bees and foragers. However, other studies which mainly focused on the variation during the honey producing season observed little to no differences [19, 20]. To our knowledge, however, there is no large-scale study to evaluate the effect of environment on the honey bee gut bacteriome. Although previous studies have found differences between honey bees kept in two different locations [12, 17], detailed understanding of environmental conditions is still missing. In keeping with our initial expectations, we observed significant differences between seasonal states and environmental conditions. The $\beta$-diversity was significantly different between March and May based on NMDS ordination (Fig 3) and March samples between warmer and cooler regions differed either in their $\alpha$- and $\beta$-diversity (Figs 2 and 4). Besides, several bacterial species of the core bacteriome of our study have shown significant differences between seasonal and environmental states. Based on the NMDS ordination, less variability in $\beta$-diversity was found between apiaries in March than in May (Fig 4). Although warmer and cooler regions separated either in March or May samples as well, the observed higher variability could be a reason why this difference was found to be significant only in March (Fig 4). However, between precipitation levels, $\beta$-diversity didn't show such clear differences neither in March nor in May (Fig 4). We could explain these differences as a transition from early after winter, when bees still need feed supplementation and their

bacteriome is not fully transitioned to the summer state, to a bacteriome characteristic to summer, where apiary level and regional differences can shape its composition. Bacteriomes in the winter can show substantial differences in different insect species [56, 57]. However, during the honey-producing season, a more even distribution of species can be observed in foragers, with Gilliamella being one of the most abundant members. Although the bacterial composition of our March samples is more similar to the summer state of the honey bee gut bacteriome (indicating that it is almost completely transformed from the winter state), Lactobacilli still occupy a large proportion of the core bacteriome, which is significantly reduced in May (Fig 5). Besides, in contrast to the findings of Kešnerová and colleagues [22], *B. apis* shows a notable increase from March to May in our results (Fig 5). We could reason that the observed smaller variability of *β*-diversity in March samples (Fig 4) could be the consequence of the lack of flowering plants and the fact that beekeepers use similar feed supplements to complete the nutritional needs of the colonies. However, as the *β*-diversity of warmer and cooler LAUs significantly differed in March, it is straightforward to assume that flowering is initiated earlier in warmer regions. The elevated abundance of *L. kunkeei* could also indicate the onset of nectar collection as this is a highly specialised bacterial inhabitant of the honey crop. It was shown that *L. kunkeei* is nearly absent in winter, however its abundance gradually increases from spring to summer [58, 59]. In May, however, no significant difference in *β*-diversity was found between environmental conditions, although this could be the reason for the higher variability found between apiaries. During the peak of the honey producing season, many factors could affect the microbial composition of the gut and thus account for the higher variation. It is well known that diet can shape the microbiome composition of humans and other species [49, 60–65], including honey bees and bumblebees [66, 67]. Although owners of the apiaries were consistent in keeping their bees in regions covered in acacia, slight differences might occur between individual regions. Pesticides can also influence the gut microbial communities of honey bees [68]. Even though the owners of the apiaries sampled in this study didn't observe any sign of poisoning on their farms, subclinical pesticide exposure can affect the gut microbiome of honey bees even without any visible impact on the colony [69]. Furthermore, it is possible that intrinsic effects, such as genotype, can also affect the microbial composition of the honey bee gut, as was found in the case of Drosophyla melanogaster [70].

The honey bee is an important pollinator species worldwide [71, 72]. Its high economic value resides in its role in crop pollination and the wide variety of products they make [1–3]. They were also found to be a useful model animal for several biological research areas, including microbiome research [73, 74]. The gut bacteriome of honey bees is an essential determinant of their health. It possesses a myriad of functions that benefit its host, for example, it enables the degradation of different polysaccharides originating from the bees diet, such as pollen walls [75, 76]. It might also have a role in recycling the nitrogen waste materials of honey bee nitrogen metabolism [76] and can metabolise potentially toxic sugars for bees [75]. Besides these metabolic functions, the gut microbiome has a positive impact on the host immune system [45] and it protects the bees from different pathogens [45, 77]. Understanding the normal composition and natural variation of the gut bacteriome of honey bees is an important foundation for future research. It is necessary for understanding how different pathologic conditions can alter its composition and to work out protocols to return it to a healthy state [78]. Our results provide data on the association of the honey bee bacteriome with season, precipitation and temperature in temperate climatic conditions. Due to the two-level pooling, we can assume that the effect of other untreated factors (e.g. age heterogeneity) besides the environmental factors under investigation can be neglected. Including such strata in the sampling design of further studies would be valuable. The results presented, together with potential

future studies, can increase our understanding of the natural fluctuation of the healthy bacteriota of honey bees and could help in the preservation of their health.

## Conclusion

Based on our results, one may conclude that the composition of healthy core bacteriomes in honey bees varies depending on the climatic and seasonal conditions. This is probably since climatic characteristics and vegetation states determine the availability and nutrient content of flowering plants. The results of our study prove that in order to gain a thorough understanding of a microbiome's natural diversity, we need to obtain the necessary information from extreme ranges of the host's healthy state.

## Supporting information

**S1 Fig. Relative abundance distributions.** Boxplots denoting the actual point distribution for the differential species in comparisons were significant.
(PDF)

**S1 Table. Sample sizes by apiaries and families for the both sampling periods.** During the sampling in May, in two apiaries (ID: 6, 13) only two of the three colonies from the March sampling period were accessible so only these two were sampled.
(PDF)

## Acknowledgments

In memory of Rajnald András Köveshegyi OCist. We would like to say thanks to the beekeepers for giving us their indispensable help.

## Author Contributions

**Conceptualization:** László Békési, Róbert Farkas, László Makrai, Norbert Solymosi.

**Data curation:** László Békési, Róbert Farkas, László Makrai, Norbert Solymosi.

**Formal analysis:** Márton Papp, Norbert Solymosi.

**Writing – original draft:** Márton Papp, Norbert Solymosi.

**Writing – review & editing:** Márton Papp, László Békési, Róbert Farkas, László Makrai, Maura Fiona Judge, Gergely Maróti, Dóra Tőzsér, Norbert Solymosi.

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
