## [Decision Letter · Decision Letter 0]

14 Jun 2022

PONE-D-22-13087Natural diversity of the honey bee (Apis mellifera) gut bacteriome in various climatic and seasonal

statesPLOS ONE

Dear Dr. Solymosi,

Thank you for submitting your manuscript to PLOS ONE. After careful consideration and review by two of your colleagues, we feel that it has some merit but does not fully meet PLOS ONE’s publication criteria as it currently stands. Therefore, we invite you to submit a revised version of the manuscript that addresses the points raised during the review process. Both reviewers raise valid concerns about the content (for examples the method concern about the age or physiological state of the sampled bees) and the presentation (please correct all formatting and language issues) that need to be addressed.

We look forward to receiving your revised manuscript.

Kind regards,

Olav Rueppell

Academic Editor

PLOS ONE

Journal Requirements:

2. Please update your submission to use the PLOS LaTeX template. The template and more information on our requirements for LaTeX submissions can be found at http://journals.plos.org/plosone/s/latex

"The project is supported (MP) by the European Union and co-financed by the European Social Fund (No. EFOP-3.6.3-VEKOP-16-2017-00005). It has also received funding (NS) from the European Union’s Horizon 2020 research and innovation program under Grant Agreement No. 874735 (VEO). GM received support from the Hungarian Academy of Sciences through the Lendület-Programme (LP2020-5/2020). "

4.Thank you for stating the following in your Competing Interests section:  

"NO authors have competing interests"

Reviewers' comments:

Reviewer's Responses to Questions

**Comments to the Author**

1. Is the manuscript technically sound, and do the data support the conclusions?

Reviewer #1: Yes

Reviewer #2: Yes

2. Has the statistical analysis been performed appropriately and rigorously? 

Reviewer #1: Yes

Reviewer #2: Yes

3. Have the authors made all data underlying the findings in their manuscript fully available?

Reviewer #1: Yes

Reviewer #2: No

4. Is the manuscript presented in an intelligible fashion and written in standard English?

Reviewer #1: Yes

Reviewer #2: Yes

5. Review Comments to the Author

Reviewer #1: 1. The author does not explain in the “Introduction” why the gut bacteriome of bees should be reasearch. Although the authors mention in the abstract that "The composition of their bacteriota, for which the available knowledge is limited, is essential for their body’s functioning." , but the necessary clarifications are also required in the“Introduction” .

2. L30 "Nevertheless,......" has little to do with the previous sentence and does not understand what the author wants to express.

3. In the "Sampling design and sample collection" section of the method, the author describes it in great detail and length, it is recommended to use a condensed language to describe it, and put more detailed methods in supplementary information.

4. In the results section, it is recommended to add subheadings to increase readability.

5. It is recommended that the author add a summary to the manuscript so that the reader can better understand the author's ideas throughout the text.

6. There are many reference format errors, such as:Species names in the title need to be italicized: Articles 11, 14, 15, etc; Journal names are not uniform: Articles 18, 24, etc.........Authors are advised to carefully revise the reference format.

7.Table 1 and Table 2: Bacterial species names in the table require italics.

8. Lines 83 20-20 are incorrectly written. Lines 103 10-10 are incorrectly written.

Line 230 number italics.

9.Fig2 in line 164 should be Figure 2, and the full text should be consistent, as should Fig3-4 on line 175. Same as Fig5 in line 189. The Figures in line 256 should be changed to Figure.

Reviewer #2: Authors have performed a study focused on changes of microbiota between March ana May, moreover, they also included templerature and precipitation in their model to get insight into how these influence bacteriome. The study is of some interest, however, design, sampling protocol and statistics, especially validation need to be improved.

l. 83, 103. by n-dash you denote pairs used, but it arather looks like a range with a typo, please improve.

Please check journal requirements for in vitro /in vivo typesetting and binomial names in tables

l. 48-158 is partially repetitive, I dont think this belongs to Results section

l. 177 - please provide validation data for Permanova, low p-value itself is not accurate representation of the test.

The experimental design description needs improvement, it is not clear how many samples were used in total for analysis, also owing to the migration, a scheme would help

I recommend showing also the boxplots denoting the actual point distribution for the differential species in comparisons including GDD and precipitation, for those species that were significant.

My biggest issue is that authors do not state how the bees were sampled, there is a huge difference between roles and age, this could also lead to a bias in the study. The bees need to be of equal age. Authors must be more detailed about the sampling

Removal of outliers is not acceptable unless they are technical errors and these need to be included in the analysis, especially in case they are pooled samples. Did you perform also technical replicates? Again, the

exact number of samples/replicates used in comparisons should be mentioned in table comparisons.

Of course there is a role of time, in March, there are still owerwintering long living bees present in the hive, however, in May during the highest nectar flow the diet completely changes.

The study has limited design but if the authors manage to do a proper statistical validation and prove that they used a strong sampling protocol, it is of some interest to scientific community. I recommend major revisions.

6. PLOS authors have the option to publish the peer review history of their article (what does this mean?). If published, this will include your full peer review and any attached files.

Reviewer #1: No

Reviewer #2: No

---

## [Author Response · Author response to Decision Letter 0]

6 Jul 2022

PONE-D-22-13087

Natural diversity of the honey bee (Apis mellifera) gut bacteriome in various climatic and seasonal states

PLOS ONE

Dear Dr. Solymosi,

Thank you for submitting your manuscript to PLOS ONE. After careful consideration and review by two of your colleagues, we feel that it has some merit but does not fully meet PLOS ONE’s publication criteria as it currently stands. Therefore, we invite you to submit a revised version of the manuscript that addresses the points raised during the review process.

Both reviewers raise valid concerns about the content (for examples the method concern about the age or physiological state of the sampled bees) and the presentation (please correct all formatting and language issues) that need to be addressed.

If applicable, we recommend that you deposit your laboratory protocols in protocols.io to enhance the reproducibility of your results. Protocols.io assigns your protocol its own identifier (DOI) so that it can be cited independently in the future. For instructions see: https://journals.plos.org/. Additionally, PLOS ONE offers an option for publishing peer-reviewed Lab Protocol articles, which describe protocols hosted on protocols.io. Read more information on sharing protocols at https://plos.org/protocols?.

Authors’ response:

Thank you for the opportunity, but we believe that the details described in the methodology section of the manuscript allow to replicate our results, and that putting them on protocols.io would be a redundancy.

We look forward to receiving your revised manuscript.

Kind regards,

Olav Rueppell

Academic Editor

PLOS ONE

Journal Requirements:

https://journals.plos.org/ and 

https://journals.plos.org/

Authors’ response:

We have checked the requirements and followed them.

2. Please update your submission to use the PLOS LaTeX template. The template and more information on our requirements for LaTeX submissions can be found at http://journals.plos.org/

Authors’ response:

The revised manuscript was edited using PLOS LaTeX template.

"The project is supported (MP) by the European Union and co-financed by the European Social Fund (No. EFOP-3.6.3-VEKOP-16-2017-

Authors’ response:

In the cover letter we have modified the paragraph by the sentence "The funders had no role in study design, data collection and analysis, decision to publish, or preparation of the manuscript.".

4.Thank you for stating the following in your Competing Interests section:  

"NO authors have competing interests"

Please complete your Competing Interests on the online submission form to state any Competing Interests. If you have no competing interests, please state "The authors have declared that no competing interests exist.", as detailed online in our guide for authors at http://journals.plos.org/ 

Authors’ response:

In the cover letter we have replaced the previous statement by the sentence "The authors have declared that no competing interests exist.".

Reviewers' comments:

Reviewer's Responses to Questions

Comments to the Author

1. Is the manuscript technically sound, and do the data support the conclusions?

Reviewer #1: Yes

Reviewer #2: Yes

2. Has the statistical analysis been performed appropriately and rigorously?

Reviewer #1: Yes

Reviewer #2: Yes

3. Have the authors made all data underlying the findings in their manuscript fully available?

Reviewer #1: Yes

Reviewer #2: No

Authors’ response:

The two reviewers differ on this issue. If Reviewer #2 believes that the sequencing and metadata of the samples that we have deposited and made public in the NCBI SRA (BioProject PRJNA685398) repository are insufficient, please help us to identify what additional data we should share about the manuscript. 

4. Is the manuscript presented in an intelligible fashion and written in standard English?

Reviewer #1: Yes

Reviewer #2: Yes

5. Review Comments to the Author

Reviewer #1: 1. The author does not explain in the “Introduction” why the gut bacteriome of bees should be reasearch. Although the authors mention in the abstract that "The composition of their bacteriota, for which the available knowledge is limited, is essential for their body’s functioning." , but the necessary clarifications are also required in the“Introduction” .

Authors’ response:

Thank you for your comments, we have amended the introduction. 

2. L30 "Nevertheless,......" has little to do with the previous sentence and does not understand what the author wants to express.

Authors’ response:

Thank you for your comments, we have reworded our message.

3. In the "Sampling design and sample collection" section of the method, the author describes it in great detail and length, it is recommended to use a condensed language to describe it, and put more detailed methods in supplementary information.

Authors’ response:

It has indeed been detailed to avoid ambiguities. In reviewing the PloS ONE template and editing instructions, we see that files can only be placed in the Supporting information section, so we would leave the methodology description as part of the main text. 

4. In the results section, it is recommended to add subheadings to increase readability.

Authors’ response:

Thank you for your suggestion, we have added headings to the subsections.

5. It is recommended that the author add a summary to the manuscript so that the reader can better understand the author's ideas throughout the text.

Authors’ response:

Following the referee's suggestion, an Author summary was added to the manuscript.

6. There are many reference format errors, such as:Species names in the title need to be italicized: Articles 11, 14, 15, etc; Journal names are not uniform: Articles 18, 24, etc.........Authors are advised to carefully revise the reference format.

Authors’ response:

Thank you for your comment. Since the editorial letter requested that the final version of the manuscript be uploaded in LaTeX format, these changes are not visible in the tracked MS Word version, but are visible in LaTeX and the generated PDF.

7.Table 1 and Table 2: Bacterial species names in the table require italics.

Authors’ response:

Thanks for your comments, we have changed the species names.

8. Lines 83 20-20 are incorrectly written. Lines 103 10-10 are incorrectly written.

Authors’ response:

Thanks to the referee for pointing out that these values are not entirely clear for the reader. In fact, the values indicated are correctly given in the text, as 20 workers from each family were collected and frozen and from these frozen individuals 10 individuals per family were used to create the pool per family from which the sequencing was done. 

Line 230 number italics.

Authors’ response:

Thank you for your comment, it has been corrected. 

9.Fig2 in line 164 should be Figure 2, and the full text should be consistent, as should Fig3-4 on line 175. Same as Fig5 in line 189. The Figures in line 256 should be 

changed to Figure.

Authors’ response:

Thank you for your comment, but according to the PLoS ONE editorial guidelines Fig should be used. 

Reviewer #2: Authors have performed a study focused on changes of microbiota between March ana May, moreover, they also included templerature and precipitation in their model to get insight into how these influence bacteriome. The study is of some interest, however, design, sampling protocol and statistics, especially validation need to be improved.

l. 83, 103. by n-dash you denote pairs used, but it arather looks like a range with a typo, please improve.

Authors’ response:

Thank you for your comment, we have modified the text.

Please check journal requirements for in vitro /in vivo typesetting and binomial names in tables

Authors’ response:

Thank you for bringing this to our attention, we have checked the manuscript from this point of view.

l. 48-158 is partially repetitive, I dont think this belongs to Results section

Authors’ response:

Thank you for your comment, if it refers to section L 148-158 and not to the section L 48-158 indicated by the referee. The repetitions have been deleted.

l. 177 - please provide validation data for Permanova, low p-value itself is not accurate representation of the test.

Authors’ response:

Thank you for your comment, we indeed did not include the results of the analysis of multivariate homogeneity of group dispersions, which was performed in parallel, and we have now corrected this.

The experimental design description needs improvement, it is not clear how many samples were used in total for analysis, also owing to the migration, a scheme would help

Authors’ response:

Thank you for your suggestion, we have included Table S1 in the Supporting information to make it easier to see the sample. sizes. The two maps (Fig 1) have also been included in the manuscript with the identification of the apiaries so that the movement can be followed.

I recommend showing also the boxplots denoting the actual point distribution for the differential species in comparisons including GDD and precipitation, for those species that were significant.

Authors’ response:

Although we believe that Fig 5 presents the requested information we have prepared a Fig S1 as Supporting information.

My biggest issue is that authors do not state how the bees were sampled, there is a huge difference between roles and age, this could also lead to a bias in the study. The bees need to be of equal age. 

Authors’ response:

We agree with the referee it would be ideal for the sampled individuals to be the same age. But the question arises, how old are they? Should the ages be matched on day or on week? Or rather, which age should be chosen. Because obviously there can be differences in gut microbiota from week to week. Of the publications we cited, only one [45] actually controlled the age of the sampled individuals. Reference 12 used an approximate age, with the caveat that it is possible that workers of other ages may have been included in the sample. The other cited papers [10,11,14,17,19,20,21,22,66,68,74] do not address this important issue in sample selection. Reference 68 deals with the issue: "Since we collected random bees from the hives, thus not controlling for age, pooling guts may have masked effects due to the potential presence of outliers, such as bees that were not exposed to the treatment." In that study, the largest pooled sample consisted of 5 guts, six times smaller than our pools. As stated in the study objective, we aimed to study workers and did not specify a specific age group. As described in the Material and Methods, we collected 20 workers per family from 3 families in the bee pools. From the 20 workers per family, the intestinal tract of 10 workers was removed. Thus, we obtained 30 guts per apiary, which were pooled and sequenced. In other words, by this two-level pooling, we tried to obtain a sample from a workers that could be considered representative of the apiary. As this issue was raised by the editor and referee in the Material and methods and in the Discussion, we addressed it.

Authors must be more detailed about the sampling.

Authors’ response:

Referee 1 has indicated that the sampling is too detailed, and we would like the editor's help in deciding which referee's suggestion to follow. If further details are needed, we would ask you to specify what further details are required. 

Removal of outliers is not acceptable unless they are technical errors and these need to be included in the analysis, especially in case they are pooled samples. 

Authors’ response:

We assume that Referee 2 is indicating to the fact that we indicated in Table 2 that we estimated values omitting outliers for two estimates, using the default settings of DESeq. In the revision, we recalculated these values, keeping the outliers, and adjusted the p-values accordingly.

Did you perform also technical replicates? 

Authors’ response:

No, we had no technical replicates.

Again, the exact number of samples/replicates used in comparisons should be mentioned in table comparisons.

Authors’ response:

All data for the samples were aggregated in Table S1 mentioned earlier.

Of course there is a role of time, in March, there are still owerwintering long living bees present in the hive, however, in May during the highest nectar flow the diet completely changes. The study has limited design but if the authors manage to do a proper statistical validation and prove that they used a strong sampling protocol, it is of some interest to scientific community. I recommend major revisions.

---

## [Decision Letter · Decision Letter 1]

17 Aug 2022

Natural diversity of the honey bee (Apis mellifera) gut bacteriome in various climatic and seasonal

states

PONE-D-22-13087R1

Dear Dr. Solymosi,

We’re pleased to inform you that your manuscript has been judged scientifically suitable for publication and will be formally accepted for publication once it meets all outstanding technical requirements.

Kind regards,

Olav Rueppell

Academic Editor

PLOS ONE

Additional Editor Comments (optional):

Reviewers' comments:

Reviewer's Responses to Questions

**Comments to the Author**

1. If the authors have adequately addressed your comments raised in a previous round of review and you feel that this manuscript is now acceptable for publication, you may indicate that here to bypass the “Comments to the Author” section, enter your conflict of interest statement in the “Confidential to Editor” section, and submit your "Accept" recommendation.

Reviewer #1: All comments have been addressed

2. Is the manuscript technically sound, and do the data support the conclusions?

Reviewer #1: Yes

3. Has the statistical analysis been performed appropriately and rigorously? 

Reviewer #1: Yes

4. Have the authors made all data underlying the findings in their manuscript fully available?

Reviewer #1: Yes

5. Is the manuscript presented in an intelligible fashion and written in standard English?

Reviewer #1: Yes

6. Review Comments to the Author

Reviewer #1: The authors might have been better of adding the pollen composition of the two seasons, which would have more clearly reflected differences in diet. In addition, the difference between the types of flowers in March and May is not large in many areas, and many areas already have a lot of flowers in March（but in this paper they are climatic types of two nutritionally extreme periods）.

7. PLOS authors have the option to publish the peer review history of their article (what does this mean?). If published, this will include your full peer review and any attached files.

Reviewer #1: No

---

## [Editor Report · Acceptance letter]

31 Aug 2022

PONE-D-22-13087R1 

Natural diversity of the honey bee *(Apis mellifera)* gut bacteriome in various climatic and seasonal states

Dear Dr. Solymosi:

I'm pleased to inform you that your manuscript has been deemed suitable for publication in PLOS ONE. Congratulations! Your manuscript is now with our production department. 

Kind regards, 

on behalf of

Dr. Olav Rueppell 

Academic Editor

PLOS ONE